

# *In silico* identification of off-target pesticidal dsRNA binding in honey bees (*Apis mellifera*)

Christina L. Mogren[1] and Jonathan Gary Lundgren[2]

[1] Plant and Environmental Protection Sciences, University of Hawaii at Manoa, Honolulu, Hawai'i, United States of America
[2] Ecdysis Foundation, Estelline, SD, USA

## ABSTRACT

**Background.** Pesticidal RNAs that silence critical gene function have great potential in pest management, but the benefits of this technology must be weighed against non-target organism risks.

**Methods.** Published studies that developed pesticidal double stranded RNAs (dsRNAs) were collated into a database. The target gene sequences for these pesticidal RNAs were determined, and the degree of similarity with sequences in the honey bee genome were evaluated statistically.

**Results.** We identified 101 insecticidal RNAs sharing high sequence similarity with genomic regions in honey bees. The likelihood that off-target sequences were similar increased with the number of nucleotides in the dsRNA molecule. The similarities of non-target genes to the pesticidal RNA was unaffected by taxonomic relatedness of the target insect to honey bees, contrary to previous assertions. Gene groups active during honey bee development had disproportionately high sequence similarity with pesticidal RNAs relative to other areas of the genome.

**Discussion.** Although sequence similarity does not itself guarantee a significant phenotypic effect in honey bees by the primary dsRNA, *in silico* screening may help to identify appropriate experimental endpoints within a risk assessment framework for pesticidal RNAi.

## INTRODUCTION

The potential to silence critical gene function in pest species has led to the proposed application of RNA interference (RNAi) as a novel class of agricultural products (*Price & Gatehouse, 2008*; *Gu & Knipple, 2013*) that target several species of economically important pests (*Baum et al., 2007*; *Maori et al., 2009*; *Desai et al., 2012*; *Hajeri et al., 2014*; *Marr et al., 2014*). These RNAi-based pesticides may be delivered to the target pest via a number of methods, including transgenic plants, diet-incorporated suspensions, and topical applications of naked or encapsulated small RNAs, which elicit post-transcriptional gene silencing following ingestion. Once ingested, the insect's cellular machinery cleaves long double stranded RNA (dsRNA) molecules into small-interfering RNAs (siRNAs) that are

Corresponding author
Jonathan Gary Lundgren,
jgl.entomology@gmail.com

19–25 nucleotides in length; these siRNAs serve as the functional unit of RNAi and govern the location of gene suppression through the degradation of complementary messenger RNA molecules (*Fire et al., 1998*; *Martinez et al., 2002*; *Vermeulen et al., 2005*). To date, this process has been investigated in the control of a number of pest groups, including parasites of medical importance, urban pests, pests and pathogens of honey bees, and agricultural pests of economic importance.

While the technology promises to be target specific (*Whyard, Singh & Wong, 2009*; *Bachman et al., 2013*), there is concern that current risk assessment frameworks for genetically modified crops are not adequate to proactively assess the risks to non-target organisms (*Romeis et al., 2008*; *Lundgren & Duan, 2013*; *FIFRA-SAP, 2014*). The risks associated with RNAi to non-target organisms include immune stimulation (*Lu & Liston, 2009*), saturation of an organism's RNAi machinery that could interfere with normal cellular processes (*Grimm, 2011*; *Flenniken & Andino, 2013*), and unintentional gene silencing. Unintentional gene silencing in non-target organisms is the primary risk posed by pesticidal RNAi; within a non-target species, this unintentional gene silencing can be due to silencing the intended gene in an unintended organism (non-target binding) or silencing a different gene with sufficient sequence similarity to the dsRNA (off-target binding) (*Lundgren & Duan, 2013*; *FIFRA-SAP, 2014*). Because pesticidal RNAi poses risks to non-target organisms that are different from other pesticides, a risk assessment framework has been proposed to proactively assess these risks using a series of steps (*FIFRA-SAP, 2014*; *Roberts et al., 2015*). Indeed, the United Nations employs the precautionary principle when conducting risk assessment of genetically modified organisms to ensure that these products do not adversely affect the environment (https://bch.cbd.int/protocol; accessed November 7, 2017).

Bioinformatic analyses that compare pesticidal RNAs to non-target genomes can help focus more extensive risk assessment procedures to predict some risks (*Heinemann, Agapito-Tenfen & Carman, 2013*). The hazard to non-target organisms should be predictable if the functional genome of a non-target organism is known, recognizing that numerous circumstances influence gene silencing even when the sequence is identical between a small RNA and the non-target genome (*Kerschen et al., 2004*). Bioinformatic analyses have thus been advocated as an initial screen of potential risks posed by RNAi (*FIFRA-SAP, 2014*; *Roberts et al., 2015*). In the present study, we used *in silico* searches to determine whether putative pesticidal dsRNAs share sequence similarities with off-target regions of the honey bee (*Apis mellifera* L.), a model non-target organism. We were specifically interested in testing the hypotheses that (1) longer dsRNAs increase the potential for off-target binding, (2) non-target silencing of the target gene is dependent on relatedness of the target and non-target species, and (3) certain gene groups in the honey bee are more prone to off-target sequence similarities with pesticidal dsRNAs.

## MATERIALS AND METHODS

### Literature review
In broad terms, our approach was to examine the literature for published pesticidal RNAs against an identified suite of pests, and search the targeted gene sequences in the pests

for similarities with regions of the honey bee genome. Published studies evaluating the effects of *pesticidal* dsRNAs were searched using the ISI Web of Knowledge database, using combinations of the search terms "pesticidal," "insecticidal," "siRNA," "dsRNA," "RNAi," and "RNA interference." See the introduction for a description of these terms. Studies were included if they evaluated the pesticidal effects of a dsRNA or siRNA and provided either the RNA sequence or primer sets that allowed the RNA sequences to be determined from the target species' genome using the NCBI genome database (http://www.ncbi.nlm.nih.gov/genome/). A total of 24 studies were included, with pesticidal qualities being evaluated for 74 dsRNAs and 21 siRNAs targeting 57 genes (Data S1). These included species of medical importance (*Hajdusek et al., 2009*; *Kwon, Park & Lee, 2013*), urban pests (*Zhou et al., 2008*; *Itakura et al., 2009*), parasites and pathogens of honey bees (*Maori et al., 2009*; *Campbell, Budge & Bowman, 2010*; *Desai et al., 2012*), agricultural pests (*Mutti et al., 2006*; *Baum et al., 2007*; *Whyard, Singh & Wong, 2009*; *Tang, Wang & Zhang, 2010*; *Choudhary & Sahi, 2011*; *Wuriyanghan, Rosa & Falk, 2011*; *Gong et al., 2013*; *Ochoa-Campuzano et al., 2013*; *Yao et al., 2013*; *Christiaens, Swevers & Smagghe, 2014*; *Chu et al., 2014*; *Han et al., 2014*; *Meng et al., 2014*; *Miyata et al., 2014*; *Yu et al., 2014*), and others (*Whyard, Singh & Wong, 2009*; *Kelkenberg et al., 2015*; *Petrick et al., 2015*).

### *In silico* sequence similarity identification

Published pesticidal dsRNAs ranged from 19 to 2500+ nucleotides in length. These were queried against the annotated honey bee genome accessed through GenBank (http://blast.ncbi.nlm.nih.gov/Blast.cgi) using the BLAST nucleotide algorithm for somewhat similar sequences (blastn). Similar genetic regions were mostly less than 25 nt long, the length expected for active siRNAs randomly generated from a dsRNA molecule. Sequence similarities of 19/21, 20/21, and 21/21 nt were tallied for each RNA against the honey bee genome, and the off-target gene name was recorded. Each off-target gene was only tallied once per dsRNA, even when that dsRNA targeted multiple locations along that gene. Sequence similarity for the target gene (non-target binding) was also recorded. Low quality proteins (as defined by NCBI) and genes of unknown function were excluded from the analysis, as were any regions with high sequence similarity that did not return any protein or gene information, such that the resultant database represents a conservative estimate of putative binding.

### Statistical analysis

Because data violated parametric assumptions, the number of off-target similarities were $\log(x+1)$ transformed and dsRNA length were log transformed to uphold assumptions for analysis with linear regression (Systat v.13.1; Systat San Jose, CA, USA). A chi-square test of independence was used to determine whether there was a significant effect of target taxa on the incidence of non-target binding in honey bees, and whether certain functional gene groups were targeted more frequently.

## RESULTS AND DISCUSSION

### dsRNA length-suppression

Each of the 74 pesticidal dsRNAs shared at least one region of perfect or high sequence similarity with the honey bee genome (average $28.6 \pm 3.32$ off-target homologies per dsRNA) (Data S1). However, none of the published pesticidal siRNAs (21 total, 19–23 nt in length) found sequence similarity within the honey bee genome at our specified level (19/21, 20/21, 21/21 nt matches), indicating that these much smaller sequences were more specific when focusing on a single non-target organism. This result was mirrored by *Li et al. (2015)*, though siRNAs are not always this benign: *Qiu, Adema & Lane (2005)* demonstrated that 5–80% of tested siRNAs resulted in off-target binding among diverse organisms.

Off-target sequence similarity increased significantly as the dsRNA increased in length (linear regression: $F_{1,100} = 623$, $P < 0.001$) (Fig. 1A), with every increase of 100 nt in the dsRNA resulting in 6 more predicted hits. This strong relationship between dsRNA length and potential off-target binding can be further demonstrated using only the genes described in *Miyata et al. (2014)*, in which the authors evaluated the effects of dsRNA length on RNAi activity *in vivo* in western corn rootworms. Although the gene targets in this study were not pesticidal specifically, and thus excluded from our overall analysis, the authors evaluated silencing of the same gene targets (*laccase 2* and *ebony*) using different sized dsRNAs to evaluate efficacy. When we examined this suite of genes from a risk assessment perspective using the same methodology as for the pesticidal RNAs, the longer dsRNAs returned significantly more regions of off-target sequence similarity in the honey bee genome (*laccase 2*: $F_{1,5} = 181$, $P < 0.001$; *ebony*: $F_{1,2} = 103$, $P = 0.01$) (Fig. 1B). While intuitive (*Bolognesi et al., 2012*), this is the first demonstration of the possibility for increased length-suppression in a non-target organism. Thus, optimizing RNA length to have maximum gene suppression efficacy in the target pest needs to be balanced against the non-target risks posed by longer molecules.

### Target-species specificity

Taxonomic relatedness of the target organism to honey bees had no effect on potential binding of siRNAs on the original gene target (non-target binding) ($\chi^2 = 9.4$, $df = 7$, $P = 0.23$) (Fig. 2). Contrary to assertions of pesticidal specificity (*Bachman et al., 2013*), this implies that silencing of the target gene in a non-target organism may be more likely to occur from random sequence similarities than based on evolutionary relatedness to the target organism. Although the pool of available literature is limited with regards to targeted applications of RNAi against pest species, with certain species being more frequently researched (e.g., *Diabrotica virgifera*), our results suggest that non-target hazard assessments should focus on species of ecological relevance rather than strictly on phylogenetic relatedness to the target species.

Unfortunately, when conducting bioinformatics analyses for the purposes of a risk assessment, the availability of sequenced genomes from representative species becomes a limiting factor. Further, the potential non-target community will differ depending on the specific pest being targeted, making it difficult to have a standard suite of species

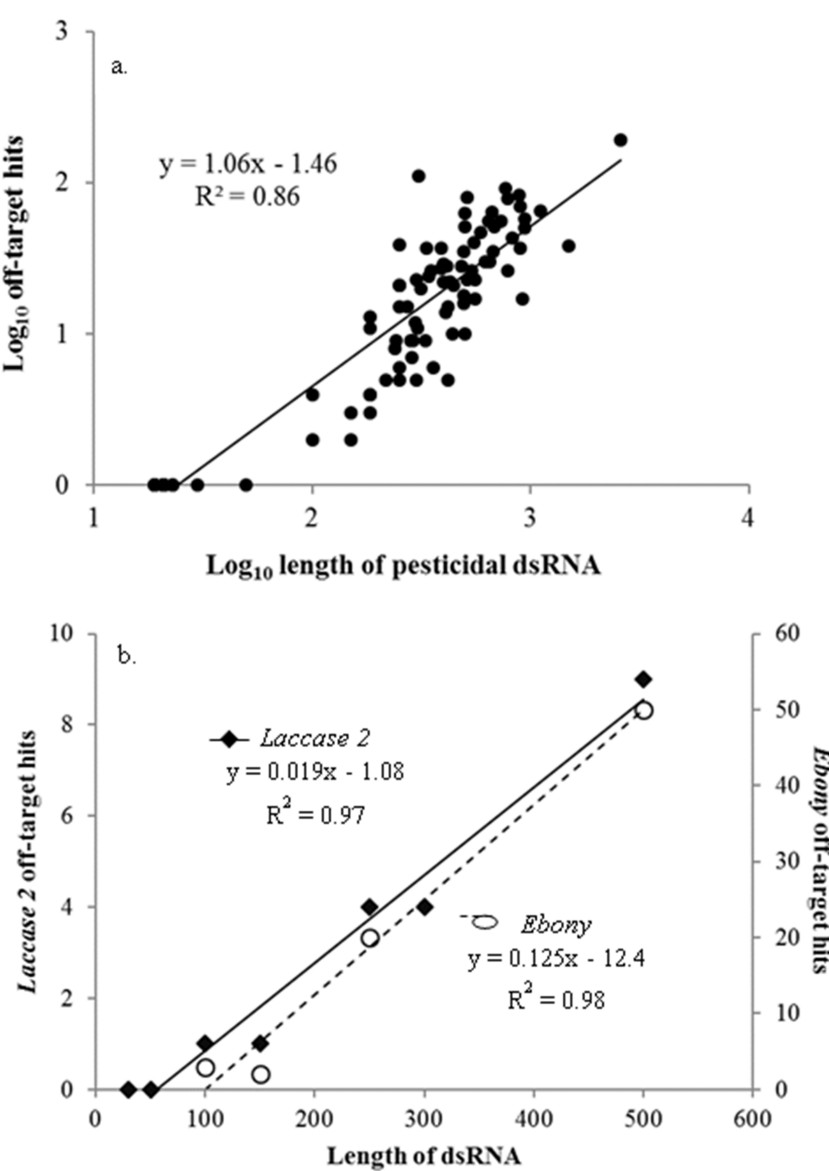

**Figure 1 Pesticidal dsRNA length and potential off-target binding in honey bees.** The relationship between pesticidal dsRNA length and potential off-target binding in honey bees for pesticidal dsRNAs (A) and the non-pesticidal *laccase 2* and *ebony* genes (data from *Miyata et al. (2014)*) (B).

to evaluate for non-target effects. Bioinventories are crucial for identifying appropriate non-target species for each target pest. Supporting initiatives such as i5K (*i5K Consortium, 2013*), which strives to sequence the genomes of 5,000 representative invertebrates, and making these genomes freely available, will bolster the applicability of future *in silico* analyses aimed at identifying potential risks of gene-oriented pest control.

## Targeted gene groups

The homeobox genes and other genes involved in embryonic and developmental pathways in honey bees frequently shared sequence similarity with the pesticidal dsRNAs, particularly

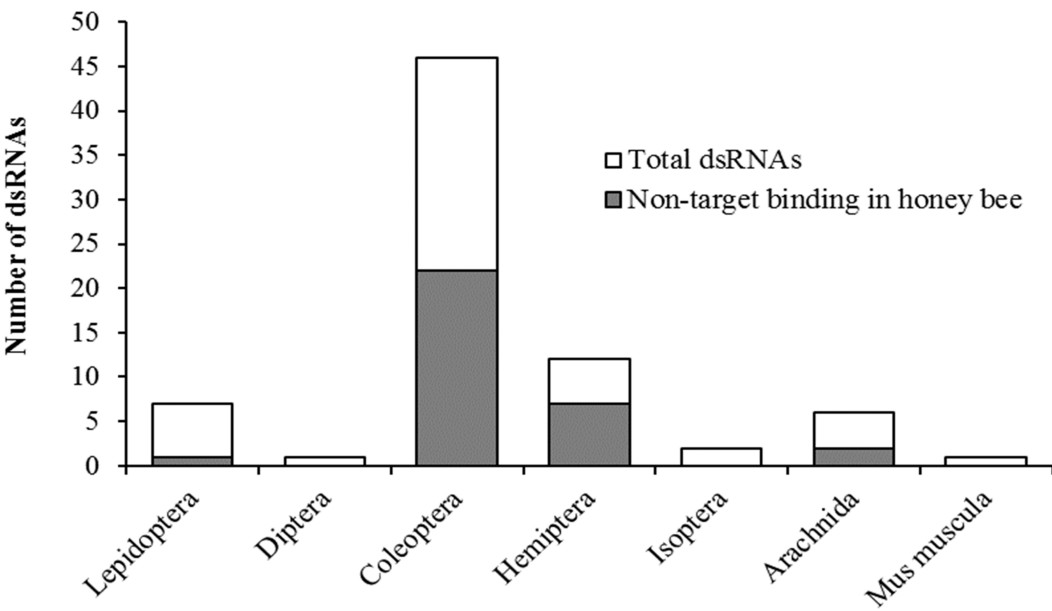

**Figure 2** **Pesticidal dsRNA target organisms and the likelihood of off-target binding in the honey bee genome.** Potential non-target binding of pesticidal dsRNAs in honey bees ($y$-axis, shaded area) versus the original target taxa ($x$-axis), in relation to the total number of examined pesticidal dsRNAs. Taxa are ordered by increasing relative divergence time from honey bees.

when vATPase subunits were the pesticidal targets ($\chi^2 = 10$, $df = 4$, $P = 0.03$). 67% of all tested dsRNAs had the high potential to bind to honey bee developmental genes that were not the target, and 33% of these shared high similarity with homeobox genes specifically (Data S1). Although we have an incomplete picture of which genes are expressed in most genomes at any given time, many of these genes, while important during embryogenesis and development, support additional critical functions such as cell proliferation and apoptosis, and are highly conserved across metazoans. In this instance, *in silico* analysis identified some of the potential gene targets that could present a hazard requiring unique assessments across life stages to properly identify a phenotypic effect. If validated in future *in vivo* assessments, this screening method may prove useful in identifying appropriate experimental endpoints in non-target risk assessments.

## CONCLUSIONS

Our bioinformatics-based *in silico* analysis provides a conservative assessment of potential off-target binding of pesticidal dsRNAs in the honey bee genome; the actual binding affinity of RISC is more nuanced than 100% or similar sequence similarity for subsequent mRNA degradation. While some have documented suppression of off-target gene expression with 20/21 nt similarity (*Jarosch & Moritz, 2012*), others have found silencing with even less sequence similarity in certain study systems, particularly in the 2–8 nt seed region of the siRNA. For example, in experiments with cultured human cells, *Saxena, Jonsson & Dutta (2003)* found gene silencing with as many as 3–4 bp mismatches in addition to G.U wobbles

(guanine and uracil have a slight affinity for each other), while *Jackson et al. (2003)* found mRNA degradation with only 11/21 contiguous nt. The locations of the mismatches along the siRNA are also important; perfect sequence similarity of the seed region is particularly crucial for mRNA recognition (*Jackson et al., 2006*; *Chu et al., 2014*). Additionally, the off-target effects may not be easily predicted, as downstream effects may result in gene suppression unrelated to the sequence of the original pesticidal molecule (*Hanning et al., 2013*). Another area of research that merits attention from a risk assessment perspective is that of RNA replication of the primary pesticidal molecule within a non-target organism. It has been repeatedly shown that environmental RNAi is replicated within a cell, and that the secondary RNAs produced do not always perfectly match the original sequence of the insecticidal RNA (*Pak & Fire, 2007*; *Sijen et al., 2007*). Focusing *in silico* analyses on only the primary insecticidal molecule may overlook these potential non-target effects.

However, *in silico* identification of sequence similarity between a pesticidal dsRNA and non-target organism's genome does not imply that RNAi will occur in the non-target organism. A fundamental difference between RNAi and chemical pesticides resides in their spectrum and mode of activity. Arguably, biochemical pesticides work on a limited number of physiological targets within an organism, and the list of potential non-target species is restricted to those sharing these targets with the pest. The absence of a relationship of taxonomic relatedness of target and off-target species and the likelihood of gene similarity between them indicates that the list of species potentially at risk from RNAi initially includes all of those species that use mRNA for gene expression and have the cellular machinery to process small RNAs. This spectrum of activity, and broad set of potential unintended phenotypic effects of the pesticidal RNAi may make predicting the risk of this technology more challenging than other pesticides. Unintended gene silencing will depend on a number of factors. The organism would need to possess behavioral characteristics that would put it into contact with contaminated materials, e.g., leaf tissue *versus* pollen *versus* nectar feeding at a contaminated location. Other factors include the length of the dsRNA and whether the organism is exposed to siRNA or dsRNA, the identity of the target or off-target mRNA, the size of a non-target organism's genome (more off-target binding would be expected when there are more potential gene targets), the necessary binding affinity of a particular siRNA, exposure concentration of the dsRNA, and the physiological state of the insect (*Qiu, Adema & Lane, 2005*; *Baum et al., 2007*; *Huvenne & Smagghe, 2010*; *Gu et al., 2014*).

Ecological risk assessment is a complex and multi-stepped process, and no single piece of work is sufficient to fully quantify the risk of a toxicological event. We have demonstrated that an *in silico* analysis may be used as a first step in establishing whether off-target binding could pose a significant threat for a particular pesticidal dsRNA in a non-target organism such as the honey bee. Future experiments to evaluate the usefulness of this tool are planned that would quantify up/down gene regulation of honey bees exposed to pesticidal dsRNA. Taken together, these data may provide a basis for designing biologically appropriate experiments to optimize hazard assessments for applications of this novel pesticidal technology in field settings where honey bees and other non-target organisms may be exposed.

## ACKNOWLEDGEMENTS

We thank Casey Snyder and Nathan Koens for assisting in compiling the RNAi database. Mention of trade names or commercial products is solely for the purpose of providing specific information and does not imply recommendation or endorsement by the U.S. Department of Agriculture.

### Funding

This work was funded by NIFA BRAG Award SDW-2012-01639 and the USDA-ARS. The funders had no role in study design, data collection and analysis, decision to publish, or preparation of the manuscript.

### Grant Disclosures

The following grant information was disclosed by the authors:
NIFA BRAG Award: SDW-2012-01639.
USDA-ARS.

### Competing Interests

The research was funded through public funding, and neither author has patents on RNAi-based pesticides nor affiliations with entities that have interest in RNAi-based pesticide technology. Jonathan Lundgren is the director of Ecdysis Foundation.

### Author Contributions

- Christina L. Mogren conceived and designed the experiments, performed the experiments, analyzed the data, wrote the paper, prepared figures and/or tables, reviewed drafts of the paper.
- Jonathan Gary Lundgren conceived and designed the experiments, analyzed the data, contributed reagents/materials/analysis tools, wrote the paper, prepared figures and/or tables, reviewed drafts of the paper.

### Data Availability

The raw data has been uploaded as a Supplemental File.

### Supplemental Information

Supplemental information for this article can be found online at http://dx.doi.org/10.7717/peerj.4131#supplemental-information.

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
