# Peer review of "In silico identification of off-target pesticidal dsRNA binding in honey bees (Apis mellifera)"

_PeerJ, doi:10.7717/peerj.4131_

## Round 0.1 · original submission · Minor Revisions

The article provides important information regarding a novel class of agricultural products with broad application but with a potential risk for non target insects (in this case the honey bee).

I think that even if this paper is challenging and extremely interesting not only for the scientific community but also for different stakeholders (e.g. beekeepers, farmers, regulatory authorities, etc.), it needs some revisions prior to publication.

Indeed, there are substantial problems in the scientific language/terms used by the authors which cannot render the article fully clear (e.g. siRNA vs dsRNA vs gene knockdown; homology vs similarity, etc.).
I personally think that rigour is mandatory especially in scientific terminology.

I also think that the suggestions made by reviewer 1 may significantly improve the discussion.

Furthermore, as suggested by reviewer 2, given the broad interest of this new biotechnology that goes beyond US borders, I strongly encourage the authors to also consider, for example, EU regulatory risk assessment.

As well, the authors might consider to briefly discuss the “taxonomic relatedness” issue, which may provide food for thoughts to the scientific community.

Given that in my opinion the above mentioned suggestions can be addressed without excessive effort, my decision is minor revisions.

Reviewer 1 ·

Basic reporting

The article could be significantly improved by reviewing the use of jargon and revising unclear sentences.
A few additional references are recommended.

Experimental design

The authors state in the materials section that search terms included “siRNA,” “dsRNA,” “RNAi,” and “RNA interference.” In the Conclusion, they say outcome can matter depending on "whether the organism is exposed to siRNA or dsRNA". This implies that there is a meaningful distinction that is not made clear in either use of terms or methodology.

Validity of the findings

With clarification of the issue above, I have no further comment.

Additional comments

The paper uses a lot of jargon and that also makes some sentences difficult to understand even for a reader familiar with the jargon.

Please reconsider using the word homology or its derivatives (eg, homologies) when referring to quantitative relationships between DNA sequences. Homology is a conclusion of relatedness by descent and not a synonym of similarity nor a quantitative term (Fitch WM. Homology a personal view on some of the problems. Trends Genet. 2000;16:227-31). There is no ‘degree of sequence homologies’ just as there is no ‘degree of pregnancy’ and no information is lost saying ‘degree of sequence similarity’ instead. Sequences may be similar because they are homologous, or similar because they are similar no matter their histories (e.g “random sequence similarities” (line 156-7)). It doesn’t matter why they are similar for this manuscript only that they are. I know that others do it, but such uses are eroding the precision of the term.

In fact, given that taxonomy is not a predictor of off-target effects, the use of the word ‘homology’ is particularly troublesome.

33-35 This sentence could be expressed as: “The target gene sequences for these pesticidal RNAs were determined, and the degree of similarity with sequences in the honey bee genome were evaluated statistically.”

37-8 “The likelihood of off-target sequence homology increased with the parent dsRNA length.” Please reconsider this jargon. I suggest “The likelihood that off-target sequences were similar increased with the number of nucleotides in the dsRNA molecule.” What after all is the parent dsRNA (also line 137)? Is it the active pesticidal RNA?

38-9 The dsRNA doesn’t bind to the gene (as in DNA). Instead: “Non-target binding of the dsRNA was unaffected by taxonomic relatedness of the target insect to honey bees…”

42-43 Sequence similarity is not enough to predict biological effect, but likewise there are no bioinformatics ways to ensure that a dsRNA will have no effect, too. E.g. because of secondary dsRNA generation in exposed insects, the primary dsRNA may not be the active molecule in the honey bee. This sentence seems to imply otherwise.

57-59 This sentence describes two methods of delivery but likely readers will be unfamiliar with both, especially since to my knowledge there are no commercially available sprays. Consider introducing appropriate references for both.

59-63 This is true if the dsRNA is longer than the active molecule. It does not have to be. It is possible that plant processed or encapsulated dsRNAs may already be 19-25 nucleotides and not require further processing in the insect.

73-74 This is a grammatically challenged sentence. Gene sequences aren’t silenced, genes are. Please revise to something like: “within a non-target species, this unintentional gene silencing can be due to silencing the intended gene in the unintended organism (non-target binding) or silencing a different gene with sufficient sequence similarity to the dsRNA (off-target binding)”

81 What does “even when sequence homology is identical” mean? Does it simply mean ‘even when sequences are identical’?

79-82 This sentence contradicts many reports in the literature that are more recent than 2004. E.g. Hanning JE, Saini HK, Murray MJ, van Dongen S, Davis MPA, Barker EM, et al. Lack of correlation between predicted and actual off-target effects of short-interfering RNAs targeting the human papillomavirus type 16 E7 oncogene. Br J Cancer. 2013;108:450-60.
Unless the authors have other references, this sentence should be removed. The reason bioinformatics analyses have been advocated is not because (82-83) they can avoid false negatives but because they are a starting point to test putative positives. Heinemann JA, Agapito-Tenfen SZ, Carman JA. A comparative evaluation of the regulation of GM crops or products containing dsRNA and suggested improvements to risk assessments. Environment International. 2013;55:43-55.

160-1 Well said. Another good reason to use homology correctly. Moreover, with sprays, non target organisms don’t have to be animals.

162-170 This paragraph is incomplete without discussing secondary dsRNA products. Pak J, Fire A. Distinct populations of primary and secondary effectors during RNAi in C. elegans. Science. 2007;315(5809):241-4. Sijen T, Steiner FA, Thijssen KL, Plasterk RHA. Secondary siRNAs result from unprimed RNA synthesis and form a distinct class. Science. 2007;315(5809):244-7.

Please avoid associating predicted binding with actual binding, as in (line 174-5) “67% of all tested dsRNAs had off-target binding with developmental genes in honey bees” This is more accurately “had the potential bind to honey bee developmental genes that were not the target”. In fact, binding is not the key criterion because one base pair is enough to satisfy binding. Some sense of the strength of the binding should also be imparted to the sentence.

178-9 It is not correct to say that a gene ‘performs’ cell proliferation or apoptosis.

180-2 Yes, but in saying that “in silico analysis identified potential gene targets that could present a hazard” it is important to also say that it is not possible to confirm that all or even most potential unintended targets were identified. This is the first line of the conclusion section, but that is too subtle.

189-192 RNAi is not just ‘gene knockdown’. There are 3 manifestations from RNA degradation (involving siRNAs that have 20-21 nt matches usually in the coding region) to translation inhibition (normally when miRNAs are used, ie, those with 2-8nt in the seed region) and RNA directed methylation. The discussion perhaps would benefit from spelling this out.

191-197 Hanning et al (above) would be a good reference for this paragraph.

200-2. Not if it is a spray.

202-3 What is the difference between what you mean by siRNA and what you mean by dsRNA? The latter is usually the generic for si/mi/shRNA etc. Do you mean processed dsRNAs when you say siRNA? Are you talking about the guide strand?

203-7 And secondary dsRNAs.

Reviewer 2 ·

Basic reporting

GENERAL
The paper fulfills all criteria for professional scientific publication: clear and well written, literature references and structure of paper is professional.

This paper offers important information for risk assessment of an emerging new biotechnology with broad application but also risk potential for insects. The authors carried out database research for published sequence information to determine the degree of homology between target gene sequences of pesticidal active RNAs in a nontarget organism, the honey bee. Honey bees served as an iconic case example of nontarget species with critical socio-economic and ecological functions. Off-target effects of any pesticidal substance must be avoided. The authors tested and validated a new methodology for improving the current risk assessment for nontarget effects that are not geared to assess these new biotechnologies. Therefore, I consider this work very important and recommend publication.

The research method is well described and carried out, the results are very interesting clearly suggesting the potential for nontarget effects in honey bees to occur if they come into contact with the RNAi. Whether or not that will happen depends on the form of delivery of the RNAi - in-planta or as sprayable formulation - and on timing of the application. But the first step is to determine whether there are any sequence homologies and the authors clearly determined that this is the case. Hence, now exposure assessments must follow and should be carried out prior for assessing the full risk potential. All of this definitely merits publication.

My further comments address issues that are not critical to the decision of publication but are for the authors to decide to include if they agree with me that it will improve the paper. Since page numbers are missing I can only indicate line numbers.

INTRODUCTION
Line 67. The authors mention here 'current risk assessment framework' - personally I would appreciate if the authors would not only focus on the US but also include references to the Cartena Protocol Risk Assessment requirements or EU regulatory risk assessment (mandatory for over 160 nations in the world). The US is a singular case having the most lax standards re risk assessment of biotechnology. This might be used as an excuse to reject the applicability of this work/paper to other regulatory systems of biotechnology as being more stringent. However, none of the other regulatory risk assessment schemes in the world would fully caputure these new risks and, therefore, this work is definitely also of relevance beyond the US.

Line 76, last word 'unique' - I suggest the term 'different' instead - and again, also here, reference to other regulatory systems are highly recommendable, so this work has relevance beyond the US.

MATERIALS AND METHODS - see below in next section

RESULTS AND DISCUSSION
Line 153 onward - relating to 'taxonomic relatedness' as an indicator or a criteria for selecting potential nontarget species at risk. This issue drops a bit out of the air, although there is a well-known precedent. This argument for selecting potentially affected nontarget organisms for testing based on their relatedness to the target pest species as indication for susceptibility has long history in the risk assessment discourse of GM Bt plants. And there it has already or also been shown to be a weak indicator for selecting potentially affected nontarget organisms.

In general, I find the authors could connect their discussion and paper in general to the on-going discourse about the already existing GM crop plants. Some issues are indeed the same or connect in other ways. And there are several references that could be cited here in support.

One issue that is still puzzling to me is on the other hand the fundamental difference in these pest control strategies that have not been properly pointed out - at least to my knowledge which is admittedly limited. It is in my view a fundamental difference if a sprayable pesticidal substance is a low molecular weight chemical or protein (in case of Bt toxins) affecting biochemical metabolic processes or life support systems such as nervous signal transmission or breathing apparatus of insects versus this form of pest control where it targets the genetic machinery. I think we have to think about specificity in completely different way whether it targets biochemical pathways or gene functioning. In particular, in my view, this is argument to broadly reject the above mentioned issue of 'taxonomic relatedness' to be in any way indicative of susceptibility. All organisms share a very similar genetic machinery - hence, all organisms are potential nontargets in a much more broader way than it could be argued with a chemical disrupting certain biochemical processes that are unique to say a kingdom of organism or other fundamental categories of organisms (e.g. photosynthesis in plants, breathing in insects vs mammals, aquatic vs terrestrial organsims etc.). I find that this paper may be a good one to at least raise these issues or point to references where these have been discussed - if at all.

The authors are approaching these fundamental questions a little bit in an ad-hoc fashion in the last paragraph of their discussion (before conclusions) within the discussion about what are highly conserved genetic regions in metazoans or not. I think, here a more fundamental discussion along the lines I outlined above would be quite suitable and add relevance of this paper for the general discussion about risk assessment of types of biotechnology applications.

Experimental design

MATERIALS AND METHODS
All is well described but I think the process could be explained better.

Lines 92-94 - the authors list here the search terms for their database search. Did they not include the term 'honey bee' or 'Apis mellifera' or terms refering to target- or nontarget organisms? Later on they list the various species they found published studies for. This means they only looked for anything published on RNAs for pest control and then assessed as part of their analyses which the target species were? And then they set out to compare these (i.e. all RNA applications they found) against the published genome sequences of the honey bees? This process should be explained a bit better, in particular, for readers who are not familiar with this pest control technology.

Validity of the findings

Data is robust and well presented and highly relevant to the field of risk assessment of biotechnological applications and products.

Conclusions are supported by the data shown and linked to the objectives and research questions. In fact, I suggest that the authors broaden their discussion and connect it to an international discussion of regulatory risk assessment requirements to make it clearly relevant beyond the US.

---

## Round 0.2 · accepted · Accept

The authors met all the suggestions made by the reviewers and the article is now ready to be published.

Congratulations for this challenging contribution on risk assessment of RNAi-based technologies.